# Fixed-Time Circular Impact-Time Guidance with Look Angle Constraint

**Xiangxiang Li** [1], **Wanchun Chen** [1], **Zhongyuan Chen** [1,*], **Ting Wang** [2] **and Heng Shi** [3]

1 School of Astronautics, Beihang University, Xueyuan Road, No. 37, Haidian District, Beijing 100191, China; lixiangxiang@buaa.edu.cn (X.L.); wanchun_chen@buaa.edu.cn (W.C.)
2 Beijing Institute of Aerospace Systems Engineering, Beijing 100076, China; wangtinggood@163.com
3 Department of Precision Instrument, Tsinghua University, Beijing 100084, China; shiheng@tsinghua.edu.cn
* Correspondence: zhongyuan_buaa@163.com

**Abstract:** A fixed-time nonlinear circular guidance law that satisfies the impact time constraint is proposed. By utilizing the geometric principle that the length of a circular arc connecting the missile and the target can be analytically calculated, the exact expression of time-to-go is obtained. Thus, the impact time error can be shaped to zero, and the missile can intercept the target at the desired time, which is crucial in a salvo attack. The settling time of the impact time error is proved to be bounded by a fixed time, which does not depend on initial conditions, but is only determined by two guidance parameters. Moreover, the criteria for choosing the guidance parameters values are established analytically, rather than by trial-and-error or empirically, which can provide valuable guidelines for guidance law designers. To address the look angle constraint, deviated pure pursuit (DPP) is employed, and switching logic between guidance laws is provided. Unlike many existing impact time control guidance laws, the formulation of the one proposed is based on nonlinear engagement kinematics, and the implementation does not execute numerical calculations, which can improve the guidance accuracy and reduce computation burdens on the guidance system. A series of nonlinear simulations are implemented to verify the effectiveness of the proposed guidance law.

**Keywords:** impact time control; salvo attack; fixed-time convergence; look angle constraint





## 1. Introduction

The primary objective of a guidance law for a missile is to intercept a target with the minimum or zero miss distance. Moreover, other objectives, such as impact angle and impact time, also play an important role in realistic engagements based on different missions. In modern warfare, many important targets have been equipped with the advanced missile defense system, like the advanced surface-to-air missile system of surface facilities and the advanced close-in weapon system (CIWS) on warships. These missile defense systems can detect and destroy incoming missiles [1]. To overcome the threat of such systems, a salvo attack in which a group of missiles can reach the target as simultaneously as possible can be used to improve the survivability of each missile and saturate the defense system of the target. To realize a salvo attack, one approach, called impact time control guidance (ITCG), can be used to regulate the impact time of each missile to a common desired value.

The proportional navigation guidance (PNG) has been widely studied and extensively used due to its easy implementation and effectiveness. Moreover, it has been shown that guidance laws based on PNG can meet the constraint on impact time [1–6]. By utilizing linear kinematics and optimal control theory, the authors in [1] derived a useful impact time control law which combines the well-known PNG and the feedback of the impact time error. The work in [1] seems to be the first attempt at addressing the impact time control problem. As an extension of the work in [1], a guidance law was proposed to control both impact angle and impact time in [2]. Note that the acceleration rate command instead of the acceleration command was used as the control input in [2] to provide one additional degree

of freedom for impact time control. Compared with the result in [1], a more generalized impact time control guidance based on nonlinear kinematics is derived in [3] by using the trajectory driven by PNG, with an arbitrary navigation constant as a baseline. In [4], PNG, with a modified time-varying gain and a bias term, was presented based on the exact solution of the time to go, which was obtained by introducing the Gaussian hypergeometric functions. However, look angle constraints were not mentioned in [4]. Considering field-of-view (FOV) constraint, an impact time control guidance law was presented in [5] by combining PNG and an additional biased term of impact time error. Likewise, the research presented in [6] derived an impact time control guidance law based on the idea of virtual targets by adding two feedback terms about the range error on PNG. A 3D PNG-based impact time control guidance was proposed in [7] via introducing bias term in both pitch and yaw channels. However, most of the above PNG-based impact time control guidance laws used the linear engagement kinematics or estimated time-to-go, which may generate large errors when heading angle conflicts with small angle assumptions.

Additionally, Lyapunov-based guidance laws were shown effective in controlling the impact time. A nonlinear Lyapunov-based impact time control guidance law was derived in [8] based on the estimation of time-to-go, as used in [1,3]. In [9], the time-to-go was obtained in terms of the incomplete beta function of the initial heading error, which can be controlled by tuning a single parameter.

In addition, guidance laws based on the sliding mode control (SMC) method were also used to control the impact time. Harl and Balakrishnan developed a guidance law to control impact time and impact angle in [10] through a line-of-sight rate shaping technique and a second order sliding mode approach. The work in [10] is one of the earlier studies to solve the guidance terminal constraint problem using SMC. To avoid the singularity problem, a nonsingular sliding mode guidance is proposed in [11] for the missile to intercept the target at the desired impact time. SMC methods are also employed in [12–17] to control the impact time. Many impact time control guidance laws based on SMC methods have complicated structures, making it stressful to deal with the look angle constraint. Moreover, to satisfy impact time constraint, guidance gains or parameters are often tuned by trial and error, or by using an optimization routine, which can make on-line calculations less efficient.

Additionally, the trajectory shaping technique is also worth mentioning in controlling the impact time. The authors in [18] suggested a guidance law in which the guidance command was expressed as a polynomial function of downrange to go.

The look angle, meaning the angle between the missile velocity and the line of sight (LOS), can determine the axial velocity of the missile, and thus affects the flight time. Thus, regulating impact time by shaping the look angle profiles has attracted a lot of attention. In [19], the impact-time control problem was solved by imposing a quadratic polynomial shape and a cubic shape on the look-angle profile. The idea from [19] was also investigated under varying speed cases in [20].Then, the idea of look angle shaping was further studied in [21,22] to meet both the impact time and impact angle constraints. Besides, the range shaping technic was also shown to be effective in controlling impact time. The authors in [23] derived an impact time control guidance by expressing the range as a quartic polynomial function of time, in which the coefficients of the polynomial were determined by the boundary conditions. Later, the extension of this work was presented in [24], wherein, the range to target was formulated as a general-order polynomial in time, rather than a quartic polynomial, as used in [23]. In addition to look angle and range, line-of-sight (LOS) can also be shaped to address the impact time issue. In [25], a homing guidance law considering impact time and impact angle constraints under limited field-of-view was proposed by employing the line-of-sight shaping approach, where the reference LOS profiles were quartic polynomials in time. Han et.al. proposed a three-dimensional guidance law [26] for intercepting a maneuvering target with both impact angle and impact time constraints in which the quadratic LOS profiles in the pitch and yaw planes were suggested, respectively. Assuming that the LOS angle could be shaped as a polynomial function of range-to-go, an impact time constrained guidance law using

a range-based line-of-sight shaping strategy was proposed in [27]. Unlike in this paper, in [25–27], some parameter values can only be obtained by solving a series of nonlinear equations numerically, instead of determined analytically, which may impose burdens on the on-board computer.

For a class of guidance laws involving the impact time error, it is a challenging issue to precisely obtain an explicit formula of time-to-go. Commonly used methods to estimate time-to-going include using range over speed, using the truncation of an infinite series [1,3,4,7], or some other methods [9,18]. Fortunately, a closed-form solution of the impact time in deviated pure pursuit (DPP) can be obtained explicitly [28]. In [29], an impact time control guidance law based on DPP was proposed by shaping the dynamics of the exact impact time error. In [30], an optimal control-based guidance law was developed by establishing the relationships between the impact time, the desired look angle, and the nominal commanded acceleration, based on DPP. Considering the control loop dynamics, a decoupled approach where the desired lateral acceleration was derived using DPP was proposed in [31] to intercept a moving, but non-maneuvering, target at a pre-specified time. However, for stationary targets, the lateral acceleration command given by DPP becomes unbounded at interception.

In addition to DPP, the circular guidance (CG) can also provide an analytical solution of the impact time according to basic geometric rules. Conversely, in scenarios where the missile moves with a constant speed, if the desired time-to-go and current range to target are known, the corresponding desired look angle can be determined uniquely. In [32], by tracking the desired look angle, which was approximately obtained by solving a transcendental equation with Taylor series expansion, the trajectory could converge to a circular arc in a finite time. However, constraints on look angle were not considered in [32]. While in [33], Tsalik and Shima proposed two approaches to obtain the approximate desired look angle, one using the MATLAB curve-fitting tool, and the other iteratively using the Newton–Raphson method. Then, a PI-controller was used to eliminate look angle errors. As can be seen, both [32,33] involved numerical algorithms in calculating desired look angles.

Other than the ITCG methods mentioned above, the second approach to realize the simultaneous arrival is cooperative guidance, in which the communication network is often employed to improve the collaborative capacity between missiles [34–37]. The cooperative guidance method is not considered here.

In this paper, inspired by [29,33], CG is chosen as the baseline of the proposed guidance law, called fixed-time circular impact time guidance (FCITG). Unlike the work in [32,33], the proposed FCITG can meet the impact time constraint through shaping the dynamics of the impact time error, straightly detouring the estimation of the desired look angle or the time-to-go. Besides, the proposed FCITG has a wide range of the desired impact time when the limitation of the look angle is not considered. Further, the look angle constraint imposed by the seeker can be readily taken into account by introducing DPP, when necessary. Actually, if the seek can handle look angles greater than or equal to 90° during DPP phase, the upper limit on the desired impact time still does not exist.

For a salvo attack, to guarantee that different missiles launched from different platforms or different locations can hit the target simultaneously, the fixed-time stability is employed in designing the FCITG. Note that the fixed-time stability is different from the finite-time stability. Specifically, the settling time of fixed-time stability is globally bounded, even when the initial error tends to infinity, whereas the settling time of finite-time stability grows unboundedly when the initial error approaches infinity [38]. Motivated by [39], a fixed-time controller is developed to achieve zero miss distance at the desired impact time. More importantly, criteria for tuning the parameters in FCITG are also established in an explicit form, which can be utilized efficiently.

Compared to the existing ITCG laws in the literature, the main contributions of this paper can be encapsulated in the following key points.

(1) Compared to [1,6], the proposed FCITG is derived in nonlinear frameworks, which aids in preventing errors that arise due to linearization. Besides, the absence of small angle approximation enables FCITG to deal with large look angles.

(2) In comparison with [14,32,33], the proposed FCITG does not involve tracking desired look angles. Thus, numerical calculations of the desired look angle from the desired time-to-go can be circumvented, which improves the efficiency of the on-board computer.

(3) Unlike the finite-time stability used in [14,29], the fixed-time stability is employed in designing FCITG. Therefore, the impact time errors will converge to zero within a predefined fixed time, despite of initial conditions. This feature will stand out in a salvo attack, where each missile may have different initial states.

(4) Different from [15,33], the parameter values can be determined from the analytical criterion instead of by trial and error or on-line calculation at each time instant. Moreover, a certain combination of the values of the two parameters is applicable for many different desired impact times.

(5) Finally, the guidance proposed is of a simple form, thus leading to an easy implementation.

The rest of this paper is organized as follows. A preliminary summary of fixed-time stability is introduced in Section 2. In Section 3, a problem statement and the guidance law design are offered. In Section 4, the criteria for the selection of parameter values are established. The extension of FCITG to moving targets and the salvo attack scenario are investigated in Section 5. Simulations are carried out in Section 6 to validate the effectiveness of the proposed guidance law. Section 7 concludes this article.

Note that throughout this paper, for $x \in \mathbb{R}$ and $r \in \mathbb{R}^+$, the function $x \mapsto \lceil x \rceil^r$ is defined as $\lceil x \rceil^r \triangleq |x|^r \cdot \text{sign}(x)$, where $\text{sign}(x)$ is the sign function.

## 2. Preliminary

Prior to deriving FCITG, it is necessary to introduce some basic definitions of fixed-time stability [38]. Consider the following general nonlinear system:

$$\begin{cases} \dot{x} = f(t,x) \\ x(0) = x_0 \end{cases} \tag{1}$$

where $x \in \mathbb{R}^n$ denotes the state vector and $f(t,x) \in \mathbb{R}^n$ is a nonlinear function.

**Definition 1** ([40]). *The origin is said to be a globally finite-time stable equilibrium point for system (1) if it is globally asymptotically stable, and any solution of (1) reaches the origin at some finite time moment, i.e., there exists $T(x_0)$, such that $x(t, x_0) = 0$ for $\forall t \geq T(x_0)$, where $T(x_0)$ is the settling-time of the system (1).*

**Definition 2** ([38]). *The origin of system (1) is said to be fixed-time stable if it is globally finite-time stable, and the settling-time function $T(x_0)$ is bounded.*

## 3. Problem Statement and Guidance Law Design

In Figure 1, the planar engagement geometry between the missile and the target is shown. The X-O-Y is the inertial Cartesian reference frame. The missile $M$ has a constant speed $V_M$, and the target $T$ is stationary. Note that $r$ is the instantaneous range between the missile and the target, $\gamma_M$ is the flight path angle of the missile, and $\theta$ is the line-of-sight angle. Moreover, $\delta$ is the look angle, which is the angle between the missile velocity and the line of sight. The acceleration perpendicular to the velocity vector is represented by $a_M$.

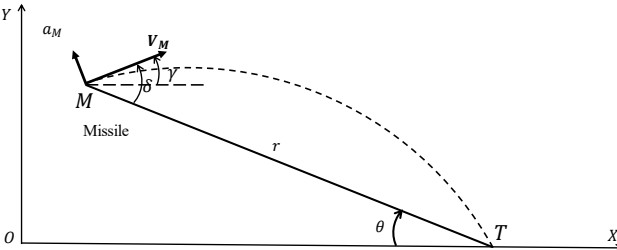

**Figure 1.** Planar engagement geometry.

The nonlinear kinematic engagement equations are given by

$$\dot{r} = -V_M \cos \delta \tag{2}$$

$$r\dot{\theta} = V_M \sin \delta \tag{3}$$

$$\dot{\gamma}_M = a_M / V_M \tag{4}$$

$$\delta = \gamma_M + \theta \tag{5}$$

Taking the derivative of (5) yields

$$\dot{\delta} = \dot{\gamma}_M + \dot{\theta} \tag{6}$$

The desired impact time is represented by $T_d$. Then the desired time-to-go is defined as

$$\bar{t}_{go} = T_d - t \tag{7}$$

where $t$ denotes the time elapsed from the beginning of the guidance.

The purpose of FCITG is to hit the target at $T_d$. If the missile keeps moving with a constant speed along a circular arc connecting its current position and the target, the total flight time can be obtained accurately and readily controlled to satisfy the impact-time requirement. In [32,33], based on the desired time-to-go, an approximation to the desired look angle can be calculated by solving a nonlinear equation. Afterwards, the look angle is regulated to converge to the desired values to eliminate impact time errors. Nevertheless, the longer the time-to-go, the greater the difference between the exact desired look angle and its approximate value. To overcome this drawback, time-to-go, rather than the look angle. is considered. For a missile guided by CG, the impact time can be obtained from geometric rules as

$$t_f = \frac{r_0 \delta_0}{V_M \sin \delta_0} \tag{8}$$

where $r_0$ and $\delta_0$ represent the initial range to target and the initial look angle, respectively.

By converting the initial values in (8) into current ones, the time-to-go under the circular guidance can be obtained as

$$\hat{t}_{go} = \frac{r\delta}{V_M \sin \delta} \tag{9}$$

Herein, $\hat{t}_{go}$ is derived without making any small angle assumptions. Note that the nonlinear framework circumvents the limitations due to linearization, especially for large initial heading errors. In (9), as $\delta \to 180°$, $\hat{t}_{go} \to \infty$, meaning that the upper limit of the intercept time is unbounded; and as $\delta \to 0°$, $\hat{t}_{go} \to r/V_M$, meaning that the lower limit of time-to-go is $r/V_M$. Thus, the feasible desired impact time is $T_d \in (T_{\min}, +\infty)$, where $T_{\min} (= r/V_M)$ corresponds to the impact time when the missile is on the direct collision course with the target. It is worth noting that the aforementioned $T_{\min}$ is not only the lower limit of the impact time of the proposed FCITG, but also of any other guidance laws.

The impact time error is defined as

$$\varepsilon = \hat{t}_{go} - \bar{t}_{go} \tag{10}$$

The main idea of FCITG is to shape the dynamics of $\varepsilon$ so that $\hat{t}_{go}$ is infinitely close to or equal to $\bar{t}_{go}$ before interception occurs. On differentiating $\hat{t}_{go}$ in (9), with respect to time, we have

$$\dot{\hat{t}}_{go} = \frac{1}{V_M}\left(\frac{\delta}{\sin\delta}\dot{r} + \frac{\sin\delta - \delta\cos\delta}{\sin^2\delta}r\dot{\delta}\right) \tag{11}$$

From (7), the time derivative of $\bar{t}_{go}$ is

$$\dot{\bar{t}}_{go} = -1 \tag{12}$$

By combining (10)–(12) the time derivative of $\varepsilon$ can be obtained as

$$\dot{\varepsilon} = \frac{1}{V_M}\left(\frac{\delta}{\sin\delta}\dot{r} + \frac{\sin\delta - \delta\cos\delta}{\sin^2\delta}r\dot{\delta}\right) + 1 \tag{13}$$

On substituting (2), (4), and (5) into (13), the dynamics of the interception time error can be derived as

$$\dot{\varepsilon} = 2 - 2\delta\cot\delta + \frac{1 - \delta\cot\delta}{V_M^2\sin\delta}ra_M \tag{14}$$

Defining

$$\begin{cases} F \triangleq 2 - 2\delta\cot\delta \\ B \triangleq \frac{1-\delta\cot\delta}{V_M^2\sin\delta}r \end{cases} \tag{15}$$

One can rewrite (14) as

$$\dot{\varepsilon} = F + Ba_M \tag{16}$$

Based on the present analysis, a main proposition is given below.

**Proposition 1.** *For a planar missile-target engagement, if lateral acceleration of the missile is chosen as*

$$a_M = -\frac{F + K\Xi}{B} \tag{17}$$

*where $\Xi \triangleq \lceil\varepsilon\rceil^{\alpha/(2-\alpha)} + \varepsilon + \lceil\varepsilon\rceil^{(4-3\alpha)/(2-\alpha)}$, K and $0 < \alpha < 1$ are two positive constants to be tuned, then $\varepsilon$ is guaranteed to converge to zero within a fixed time, only depending on K and $\alpha$ for any arbitrary initial condition. As a result, after the impact time error becomes zero, the missile will move along a circular arc and hit the target at the desired impact time, with zero miss distance.*

**Proof.** Taking the following Lyapunov function candidate into account,

$$\begin{aligned} V = {} & 2K\left(\tfrac{\alpha}{2-\alpha}+1\right)|\varepsilon|^{\frac{4-3\alpha}{2-\alpha}+1} \\ & + 2K\left(\tfrac{4-3\alpha}{2-\alpha}+1\right)|\varepsilon|^{\frac{\alpha}{2-\alpha}+1} \\ & + K\left(\tfrac{\alpha}{2-\alpha}+1\right)\left(\tfrac{4-3\alpha}{2-\alpha}+1\right)|\varepsilon|^2 \end{aligned} \tag{18}$$

The time derivative of $V$ can be obtained as

$$\dot{V} = \Gamma\left(\lceil\varepsilon\rceil^{\frac{4-3\alpha}{2-\alpha}} + \lceil\varepsilon\rceil^{\frac{\alpha}{2-\alpha}} + \lceil\varepsilon\rceil\right)\dot{\varepsilon} \tag{19}$$

where $\Gamma$ is defined as $\Gamma \triangleq 2K\left(\tfrac{\alpha}{2-\alpha}+1\right)\left(\tfrac{4-3\alpha}{2-\alpha}+1\right)$.

Substituting (17) into (16) yields

$$\dot{\varepsilon} = -K\left(\lceil\varepsilon\rceil^{\alpha/(2-\alpha)} + \varepsilon + \lceil\varepsilon\rceil^{(4-3\alpha)/(2-\alpha)}\right) \tag{20}$$

On substituting (20) into (19), one can get

$$\dot{V} = -K\Gamma\left(\lceil\varepsilon\rceil^{(4-3\alpha)/(2-\alpha)} + \lceil\varepsilon\rceil^{\alpha/(2-\alpha)} + \lceil\varepsilon\rceil\right)\left(\lceil\varepsilon\rceil^{\alpha/(2-\alpha)} + \varepsilon + \lceil\varepsilon\rceil^{(4-3\alpha)/(2-\alpha)}\right) \quad (21)$$

On expanding (21) and rearranging, $\dot{V}$ can be expressed as

$$\dot{V} = -K\Gamma\left[|\varepsilon|^2 + \left(|\varepsilon|^{(4-3\alpha)/(2-\alpha)}\right)^2 + \left(|\varepsilon|^{\alpha/(2-\alpha)}\right)^2\right] - K\Gamma\left(2|\varepsilon|^{(6-4\alpha)/(2-\alpha)}\right)$$
$$-K\Gamma\left(2|\varepsilon|^{2/(2-\alpha)}\right) - K\Gamma\left(2|\varepsilon|^2\right) \leq 0 \quad (22)$$

Note that when error $\varepsilon$ approaches to zero, system (20) can be approximated by the 0-limit subsystem [39]

$$\dot{\varepsilon} = -K\lceil\varepsilon\rceil^{\alpha/(2-\alpha)} \quad (23)$$

Then, to verify the asymptotic of system (23), a Lyapunov function is selected as

$$V_0 = K|\varepsilon|^{\frac{\alpha}{2-\alpha}+1} \quad (24)$$

The time derivative of $V_0$ can be calculated as

$$\dot{V}_0 = K\left(\frac{\alpha}{2-\alpha}+1\right)\lceil\varepsilon\rceil^{\frac{\alpha}{2-\alpha}}\dot{\varepsilon} = -K^2\left(\frac{\alpha}{2-\alpha}+1\right)|\varepsilon|^{\frac{2\alpha}{2-\alpha}} \leq 0 \quad (25)$$

While, when impact time error $\varepsilon$ approaches to infinity, system (20) can be reduced to the $\infty$-limit subsystem [39]

$$\dot{\varepsilon} = -K\lceil\varepsilon\rceil^{(4-3\alpha)/(2-\alpha)} \quad (26)$$

Like the analysis in (23), a Lyapunov function candidate of system (26) and its derivative can be obtained as

$$\begin{cases} V_\infty = K|\varepsilon|^{\frac{4-3\alpha}{2-\alpha}+1} \\ \dot{V}_\infty = -K^2\left(\frac{4-3\alpha}{2-\alpha}+1\right)|\varepsilon|^{\frac{4-3\alpha}{2-\alpha}} \leq 0 \end{cases} \quad (27)$$

Since Lyapunov function candidates are positive and their corresponding derivatives are semi-negative, systems (20), (23), and (26) are all asymptotically stable. According to Lemma 1 in [39], it can be concluded that the original system (20) is fixed-time convergent. As a result, the impact time error can converge to zero within a fixed time, independent of initial conditions. Finally, the missile will move along a circular arc toward the target.

This completes the proof. □

**Remark 1.** *Given that $0 < \alpha < 1$, it can be derived that $\frac{\alpha}{2-\alpha} < 1$ and $\frac{4-3\alpha}{2-\alpha} > 1$. Hence, a noticeable feature of system (20) is that it contains one term with a fractional exponent smaller than 1, and another one term with a degree more than 1. When $\varepsilon \gg 1$, the component using the degree larger than 1 can enable a more rapid rate of convergence. Likewise, when $\varepsilon \ll 1$, the term with the fractional exponent smaller than 1 predominates other terms to provide faster convergence. In summary, using different powers can ensure a faster convergence in any case.*

Upon substituting $F$ and $B$ from (15) into (17), the lateral command acceleration can be obtained as

$$a_M = \underbrace{-\frac{2V_M^2 \sin\delta}{r}}_{\text{Circular guidance command}} - \underbrace{K\frac{V_M^2 \sin\delta}{r(1-\delta\cot\delta)}\Xi}_{\text{Impact time error correction command}} \quad (28)$$

The first term of (28) is the circular guidance command required by the missile to stay on a circular course. The second term is to eliminate the impact time error. If the initial

impact time produced by CG is not equal to the desired impact time, this correction term could bring the missile onto a circular course to achieve the guidance objectives. Assuming $\Xi = 0$ in (28), the guidance command becomes that of circular guidance, which will be shown in the following proposition.

**Proposition 2.** *With the guidance command given by (28), once $\Xi$ becomes zero at $t = t_c$, the radius of the missile's trajectory remains constant, and retains its value at $t_c$ till the end of the engagement.*

**Proof.** At any moment, the radius of curvature of the missile trajectory can be obtained as

$$R = \frac{V_M^2}{|a_M|} \tag{29}$$

Substituting (28) into (29) yields

$$R = \frac{V_M^2}{\left| \frac{2V_M^2 \sin \delta}{r} + K \frac{V_M^2 \sin \delta}{r(1 - \delta \cot \delta)} \Xi \right|} \tag{30}$$

At $t = t_c$, the impact time error $\varepsilon$ becomes 0 as well as $\Xi$. Hence, $R$ can be written as

$$R = \frac{r}{2 \sin \delta} \tag{31}$$

The time derivative of $R$ can be obtained as

$$\dot{R} = \frac{-2V_M \cos \delta \sin \delta - r \cos \delta \frac{a_M}{V_M}}{2 \sin^2 \delta} \tag{32}$$

On substituting (28) with $\Xi = 0$ into (32), one can obtain

$$\dot{R} = 0 \tag{33}$$

which means that the missile will fly along a circular arc, with a constant radius, to the target from the time $t = t_c$ on.

This completes the proof. $\square$

To deal with look angle constraint problems, the variation of look angles will be studied. On substituting (28) into (6), $\dot{\delta}$ can be obtained as

$$\dot{\delta} = -\frac{V_M \sin \delta}{r} \left( 1 + K \frac{1}{1 - \delta \cot \delta} \Xi \right) \tag{34}$$

**Proposition 3.** *The impact time error $\varepsilon$ and $\Xi$ always have the same sign, meaning they always remain identical in positivity, negativity, or nullity.*

**Proof.** If $\varepsilon = 0$, $\Xi = 0$ can be deduced from the definition of $\Xi$ in Proposition 1. If $\varepsilon > 0$, $\Xi$ composes of three positive terms. Consequently, $\Xi > 0$ for $\varepsilon > 0$, then

$$\begin{aligned} \Xi &= -|\varepsilon|^{\alpha/(2-\alpha)} - |\varepsilon|^{(4-3\alpha)/(2-\alpha)} + \varepsilon \\ &\leq -2\sqrt{|\varepsilon|^{\alpha/(2-\alpha)} |\varepsilon|^{(4-3\alpha)/(2-\alpha)}} + \varepsilon \\ &= -2|\varepsilon| + \varepsilon < 0 \end{aligned} \tag{35}$$

This completes the proof. $\square$

**Remark 2.** *According to the proof of Proposition 1, $\varepsilon\varepsilon_0 > 0$ before $\varepsilon$ becomes zero. If $\varepsilon_0 > 0$, then $\varepsilon > 0$ as well as $\Xi > 0$ by Proposition 3. Consequently, $\dot{\delta} < 0$ holds throughout the engagement from (34). Therefore, the look angle will decrease monotonically to zero in case of $\varepsilon_0 > 0$. Further, taking the partial derivative of (9) with respect to $\delta$ yields*

$$\frac{\partial \hat{t}_{go}}{\partial \delta} = \frac{r \cos \delta}{V_M} \frac{(\tan \delta - \delta)}{\sin^2 \delta} \tag{36}$$

*It can be observed from (35) that $\frac{\partial \hat{t}_{go}}{\partial \delta} > 0$ holds for $\delta \in (0, \pi/2)$. Thus, when $\hat{t}_{go} > \bar{t}_{go}$, the look angle is expected to be smaller to reduce the impact time error.*

**Remark 3.** *Here, one of the rules for tuning K is proposed. If $\varepsilon_0 < 0$, a larger look angle is required for $\hat{t}_{go}$ to track $\bar{t}_{go}$ from (35), which means that $\dot{\delta}$ ought to be positive in the initial phase. Known from the proof of Proposition 3, $\varepsilon < 0$ produces $\Xi \leq 3\varepsilon$. Hence, the inequality*

$$1 + K\frac{1}{1 - \delta_0 \cot \delta_0}\Xi_0 \leq 1 + K\frac{1}{1 - \delta_0 \cot \delta_0}(3\varepsilon_0) \tag{37}$$

*holds at the beginning.*
*If K is chosen as*

$$K > \frac{1 - \delta_0 \cot \delta_0}{(-3\varepsilon_0)} \tag{38}$$

*then*

$$1 + K\frac{1}{1 - \delta_0 \cot \delta_0}\Xi_0 < 0 \tag{39}$$

*is ensured. Accordingly, $\dot{\delta} > 0$ can be achieved due to (34). Thus, $\delta$ will increase at the beginning of the engagement.*

Now, the behavior of $\delta$ for $\varepsilon_0 < 0$ with $K$ chosen as (36) is investigated. For the sake of convenience, we define

$$\Delta \triangleq 1 + K\frac{1}{1 - \delta \cot \delta}\Xi \tag{40}$$

assuming that $\delta$ keeps increasing before $\Xi$ vanishes. As a result, the value of $\Delta$ will always turn out to be positive because $\varepsilon$ will approach zero according to Proposition 1, and the value of $1/(1 - \delta \cot \delta)$ will decrease due to the growing value of $\delta$ under the above assumption. Consequently, $\delta$ will decrease due to (34), which contradicts the above assumption. In summary, if $\varepsilon_0 < 0$ and $K$ was chosen as stated in (36), $\delta$ will rise and then fall rather than keep rising.

Further, to study the singularity of the guidance command later, a proposition about the emergence of a zero look angle is suggested.

**Proposition 4.** *Given that $\delta_0 \neq 0$, $\delta = 0$, occurs only at the end of the engagement.*

**Proof.** It can be stated from Proposition 1 that $\varepsilon\dot{\varepsilon} < 0$ for $\varepsilon \neq 0$, which shows that the magnitude of $\varepsilon$ keeps decreasing with time till zero.

If $\varepsilon_0 > 0$, then $\varepsilon > 0$, which means that

$$\frac{r\delta}{V_M \sin \delta} - \bar{t}_{go} > 0 \tag{41}$$

holds before the impact time error becomes zero at $t_c$. If $\exists t \in (0, t_c)$, such that $\delta \to 0$, then

$$\lim_{\delta \to 0}\left(\frac{r\delta}{V_M \sin \delta} - \bar{t}_{go}\right) = \frac{r}{V_M} - \bar{t}_{go} < 0 \tag{42}$$

So far, there is a conflict between (39) and (40). So, in cases where $\varepsilon_0 > 0$, $\delta = 0$ only appears at the end of the circular trajectory at time $t = T_d$.

If $\varepsilon_0 < 0$, then $\varepsilon < 0$. As seen in Remark 3, $\delta$ will first increase and then decrease when $K$ is selected properly. In the phase where $\delta$ increases, $\delta$ evidently cannot be zero. While, in a phase where $\delta$ decreases, once $\varepsilon$ converges to zero at $t_c$, the trajectory will change to a circular arc, where the look angle will decrease to zero with a constant rate, which had been proved in [28]. That is, $\delta$ is not zero at $t = t_c$. Hence, there does not exist $t \in (0, t_c)$, such that $\delta = 0$. Finally, $\delta$ can only be zero at the end of the engagement in cases where $\varepsilon_0 < 0$.

This completes the proof. $\square$

To address the look angle constraints, we define $\delta^*$ as the allowed maximal look angle, such as $80°$, $90°$ and $100°$, which is limited by the seeker. If the desired impact time is less than the initial expected impact time by circular guidance, the look angle is monotonically decreasing, according to Remark 2. If $\delta_0 < \delta^*$ is provided, $\delta$ will not challenge the look angle constraints.

If the desired impact time is greater than the initial expected impact time by circular guidance, the look angle will first increase according to Remark 3, and then the maximum look angle represented by $\delta_{\max}$ occurs when $\Delta = 0$ or $\dot\delta = 0$. If $\delta_{\max} \leq \delta^*$, the proposed FCITG can be used throughout the engagement. However, if $\delta_{\max} > \delta^*$, to avoid violating the look angle constraint, as soon as $\delta$ reaches $\delta^*$ at time denoted by $t_1$, a deviated pure pursuit (DPP) will be applied to keep the look angle constant. On substituting instantaneous states of the missile generated by DPP into equations under FCITG, the virtual impact time error can be expressed as

$$\widetilde\varepsilon = \left(1 - \frac{\delta^* \cos\delta^*}{\sin\delta^*}\right)t + \frac{(r_1 + t_1 V_M \cos\delta^*)\delta^*}{V_M \sin\delta^*} - T_d \tag{43}$$

where $t_1$ and $r_1$ represent the time and the instantaneous range when DPP comes to be applied, respectively.

If the missile keeps flying under DPP from $t_1$ before reaching the target, the impact time can be calculated as

$$t_f^{DPP} = \frac{r_1}{V_M \cos\delta^*} + t_1 \tag{44}$$

However, in most cases, $t_f^{DPP}$ is not equal to $T_d$. Thus, it is important to determine the switching time from DPP to FCITG. Then, a useful proposition is put forward.

**Proposition 5.** $\exists t_2 \in \left[t_1, t_f^{DPP}\right)$, such that $\Delta = 0$.

**Proof.** At $t = t_1$, $\Delta \leq 0$ can be guaranteed because the look angle is increasing at that moment or has reached the relative maximal value. If $\Delta = 0$ at $t = t_1$, the proposition apparently stands. Now, the case $\Delta < 0$ at $t = t_1$ is mainly investigated.

On substituting (42) into (41) and rearranging, the final virtual impact time error can be obtained as

$$\widetilde\varepsilon_{t_f} = \widetilde\varepsilon_{t_1} + \left(1 - \frac{\delta^* \cos\delta^*}{\sin\delta^*}\right)\frac{r_1}{V_M \cos\delta^*} \tag{45}$$

In a real engagement, to meet the impact time constraint, $\delta^* \geq \cos^{-1}[r_0/(V_M T_d)]$ should hold, independent of the guidance law. The longer the desired impact time, the greater $\delta^*$ should be. If $\delta^*$ is large enough, the second term of (43) could be positively significantly large. Thus, $\widetilde\varepsilon_{tf} > 0 \to \Delta_{tf} > 0$ from (38). To be more specific, when $\varepsilon_0 > 0$, we can obtain from the proof of Proposition 1 that

$$\varepsilon_0 < \widetilde\varepsilon_{t1} < 0 \tag{46}$$

and in stage from $t = 0$ to $t = t_1$ where the look angle increases, one can get

$$r_1 > r_0 - V_M t_s \cos \delta_0 \tag{47}$$

where $t_s$ is the settling time of the impact time error, which will be solved in Section 4.

If

$$\varepsilon_0 + \left(1 - \frac{\delta^* \cos \delta^*}{\sin \delta^*}\right) \left(\frac{r_0}{V_M \cos \delta^*} - \frac{\cos \delta_0}{\cos \delta^*} t_s\right) > 0 \tag{48}$$

holds, then combining (43)–(46) yields

$$\widetilde{\varepsilon}_{tf} > 0 \rightarrow \Delta > 0 \tag{49}$$

Note that (46) is a sufficient condition for (47). During PPD, $\widetilde{\varepsilon}$ and $\Delta$ are both continuous functions of time. Therefore, $\exists t_2 \in \left(t_1, t_f^{DPP}\right)$, such that $\Delta = 0$.

This completes the proof. □

Once $\Delta$ becomes positive at $t_2$, $\dot{\delta}$ will be negative; as seen from (34), the proposed FCITG will be employed again till interception. Based on the preceding analysis, when the look angle constraint is considered, the guidance command by FCITG can be modified as

$$a_M = \begin{cases} -\frac{2V_M^2 \sin \delta}{r} - K \frac{V_M^2 \sin \delta}{r(1 - \delta \cot \delta)} \Xi, \text{if } \delta < \delta^* \text{or } \Delta \geq 0 \\ -\frac{V_M^2 \sin \delta}{r}, \text{if } \delta = \delta^* \text{and } \Delta < 0 \end{cases} \tag{50}$$

Then, the singularity of the guidance command will be discussed.

**Proposition 6.** *Given that $\delta_0$ is not equal to 0, the proposed FCITG does not suffer from singularity problems.*

**Proof.** From Proposition 4, if $\delta_0 \neq 0$, zero look angle only occurs at the interception moment meaning that $\delta$ cannot be zero before $\Xi$ converges to zero. Thus, the value of $\sin \delta / (1 - \delta \cot \delta)$ in (48) is always bounded before $\Xi$ vanishes, ensuring the boundedness of the guidance command. After $\Xi$ vanishes, the guidance command given by (28) or (48) degenerates to that of circular guidance, in which the guidance command stays constant.

This completes the proof. □

Note that even if $\delta_0 = 0$ occurs, the lateral acceleration can deviate the look angle from zero before FCITG is applied. However, when $\delta_0$ is close to zero and $\varepsilon_0 \neq 0$, the guidance command given by (48) will grow considerably large. For example, the value of $\sin \delta / (1 - \delta \cot \delta)$ will blow up to about 300 when $\delta_0 = 0.01$ rad. To address the singularity problem, we can saturate the magnitude of the guidance command to its maximal value denoted by $a_M^{max}$. In practice, the guidance command in (48) can be reformulated as

$$a_M = \begin{cases} a_M, \text{if} |a_M| \leq a_M^{max} \\ a_M^{max} \cdot \text{sign}(a_M), \text{if} |a_M| > a_M^{max} \end{cases} \tag{51}$$

## 4. The Criteria for Tuning Parameters

In this section, the criteria for the selection of the values of $K$ and $\alpha$ will be established. Before the establishment of the criteria, a proposition about the settling time of the system (20) is provided.

**Proposition 7.** *The settling time of system (20) is bounded by $\frac{\sqrt{3}}{9} \frac{2-\alpha}{1-\alpha} \frac{\pi}{K}$*

**Proof.** Although the fact that the settling time of system (20) is bounded by a fixed time has been proved in Proposition 1, the exact value of the fixed time will be given in this proposition. The proof will consider two cases, one $\varepsilon_0 > 0$ and the other $\varepsilon_0 < 0$.

Case 1: $\varepsilon_0 > 0$

From the proof of Proposition 1, $\varepsilon\varepsilon_0 > 0$ holds when $\varepsilon \neq 0$. Thus, in case of $\varepsilon_0 > 0$, $\varepsilon$ stays positive before reaching zero. Then, (20) can be rewritten as

$$\dot{\varepsilon} = -K\left(\varepsilon^{\alpha/(2-\alpha)} + \varepsilon + \varepsilon^{(4-3\alpha)/(2-\alpha)}\right) \tag{52}$$

By separating $\varepsilon$ and $t$, we have

$$\frac{1}{\varepsilon^{\alpha/(2-\alpha)} + \varepsilon + \varepsilon^{(4-3\alpha)/(2-\alpha)}}d\varepsilon = -Kdt \tag{53}$$

On defining

$$\begin{cases} \mu = \varepsilon^{\alpha/(2-\alpha)} \\ \mu_0 = \varepsilon_0^{\alpha/(2-\alpha)} \end{cases} \tag{54}$$

One can rewrite (51) as

$$\frac{2-\alpha}{2-2\alpha}\frac{1}{(\mu+1/2)^2 + 3/4}d\mu = -Kdt \tag{55}$$

Integrating (53) yields

$$\frac{2-\alpha}{1-\alpha}\frac{1}{\sqrt{3}}\tan^{-1}\left(\frac{2\mu+1}{\sqrt{3}}\right) - \frac{2-\alpha}{1-\alpha}\frac{1}{\sqrt{3}}\tan^{-1}\left(\frac{2\mu_0+1}{\sqrt{3}}\right) = -Kt \tag{56}$$

Therefore, when $\varepsilon_0 > 0$, the solution of system (20) can be obtained as

$$t = \rho\left[\tan^{-1}\left(\frac{1 + 2\varepsilon_0^{(2-2\alpha)/(2-\alpha)}}{\sqrt{3}}\right) - \tan^{-1}\left(\frac{1 + 2\varepsilon^{(2-2\alpha)/(2-\alpha)}}{\sqrt{3}}\right)\right] \tag{57}$$

where $\rho \triangleq \frac{1}{\sqrt{3}K}\frac{2-\alpha}{1-\alpha}$.

Case 2: $\varepsilon_0 < 0$

Utilizing the similar process as in case 1, when $\varepsilon_0 < 0$, the solution of system (20) can be obtained as

$$t = \rho\left[\tan^{-1}\left(\frac{1 + 2(-\varepsilon_0)^{(2-2\alpha)/(2-\alpha)}}{\sqrt{3}}\right) - \tan^{-1}\left(\frac{1 + 2(-\varepsilon)^{(2-2\alpha)/(2-\alpha)}}{\sqrt{3}}\right)\right] \tag{58}$$

Combining (55) and (56) gives

$$t = \rho\left[\tan^{-1}\left(\frac{1 + 2|\varepsilon_0|^{(2-2\alpha)/(2-\alpha)}}{\sqrt{3}}\right) - \tan^{-1}\left(\frac{1 + 2|\varepsilon|^{(2-2\alpha)/(2-\alpha)}}{\sqrt{3}}\right)\right] \tag{59}$$

Substituting $\varepsilon = 0$ into (57) the settling time represented by $t_s$ can be obtained as

$$t_s = \rho\left[\tan^{-1}\left(\frac{1 + 2|\varepsilon_0|^{(2-2\alpha)/(2-\alpha)}}{\sqrt{3}}\right) - \frac{\pi}{6}\right] \tag{60}$$

From (58) it is clear that

$$\varepsilon_0 \to \pm\infty, t_s \to \frac{\sqrt{3}}{9}\frac{2-\alpha}{1-\alpha}\frac{\pi}{K} \tag{61}$$

This completes the proof. $\square$

Evidently, (59) agrees with the property of the fixed-time stability. If $K$ was chosen as

$$K > \frac{\sqrt{3}}{9} \frac{2-\alpha}{1-\alpha} \frac{\pi}{T_d} \tag{62}$$

such that the upper limit of $t_s$ is smaller than $T_d$, $\varepsilon$ will always converge to zero within a fixed time before interception, despite the initial error. Thus, with the recalling of (36), in case of $\varepsilon_0 < 0$, the criterion for the selection of $K$ is as follows

$$K > \max\left\{ \frac{1-\delta_0 \cot \delta_0}{3|\varepsilon_0|}, \frac{\sqrt{3}}{9} \frac{2-\alpha}{1-\alpha} \frac{\pi}{T_d} \right\} \tag{63}$$

Nevertheless, when $\varepsilon_0 > 0$, only (60) is responsible for $K$. As Remark 1 has stated, $\alpha$ was chosen as $0 < \alpha < 1$ for faster convergence of the impact time error. Hence, the criteria for the selection of values of $\alpha$ and $K$ are summarized in Table 1.

**Table 1.** Criteria for the selection of values of $\alpha$ and $K$.

| Case | Parameter | Value |
|------|-----------|-------|
| $\varepsilon_0 > 0$ | $\alpha$ | $0 < \alpha < 1$ |
|  | $K$ | $K > \frac{\sqrt{3}}{9} \frac{2-\alpha}{1-\alpha} \frac{\pi}{T_d}$ |
| $\varepsilon_0 < 0$ | $\alpha$ | $0 < \alpha < 1$ |
|  | $K$ | $K > \max\left\{ \frac{1-\delta_0 \cot \delta_0}{3|\varepsilon_0|}, \frac{\sqrt{3}}{9} \frac{2-\alpha}{1-\alpha} \frac{\pi}{T_d} \right\}$ |

## 5. Extension of FCITG

### 5.1. Non-Maneuvering Moving Target

Figure 2 shows the guidance geometry for a moving target. Herein, without loss of generality, the path angle of the target can be set to 0 degree. For a slowly moving target, such as a surface ship, it is not difficult to measure its speed. In this paper, the target was assumed to move with a constant speed $V_T$.

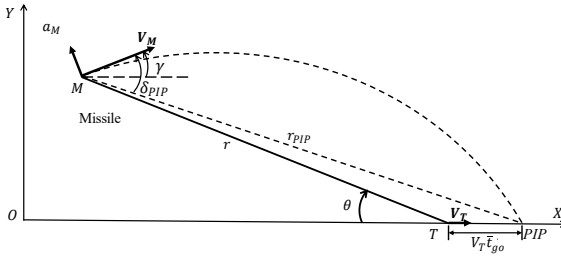

**Figure 2.** Guidance geometry for a moving target.

Similar to the method in [12], the proposed guidance law can be extended to moving targets by aiming for the predicted impact point (PIP) rather than the current position of the target. The PIP can be calculated from the known target speed and the flight-path angle.

For a moving target, by replacing the current states with corresponding states with PIP in (48), the guidance command can be obtained as

$$a_M = \begin{cases} -\frac{2V_M^2 \sin \delta_{PIP}}{r_{PIP}} - K\frac{V_M^2 \sin \delta_{PIP}}{r_{PIP}(1-\delta_{PIP} \cot \delta_{PIP})} \Xi, & \text{if } \delta < \delta^* \text{or } \Delta \geq 0 \\ -\frac{V_M^2 \sin \delta_{PIP}}{r_{PIP}}, & \text{if } \delta = \delta^* \text{and } \Delta < 0 \end{cases} \tag{64}$$

### 5.2. Salvo Attack Scenario

A salvo attack by anti-ship missiles, in which several missiles hit the target as simultaneously as possible, have been developed as one of the countermeasures against the threat of CIWS. Although each missile has a different missile-to-target range and an initial look angle, their common aim is to reach the target simultaneously. In the proposed FCITG, there exists a lower limit to the desired impact time but, no an upper limit. Therefore, to enable each missile to reach the target at $T_d$, the desired impact time of a salvo attack should be chosen as

$$T_d > \max\left\{T_i^{\min}\right\} \text{ for any } i \in \{1, 2, \ldots, n\} \tag{65}$$

where $n$ is the number of missiles involved in the salvo attack and $T_i^{\min}(= r_{0i}/V_{Mi})$ represents the initial range over the speed of the $i$-th missile.

By sharing $T_d$ and using FCITG, missiles can hit the target simultaneously. Compared with [1], firstly, the proposed FCITG does not require missiles to be launched from similar ranges, and secondly, the salvo attack desired impact time could be less than the impact time predicted by PNG for each missile. These features make the guidance law more flexible and more practical.

## 6. Numerical Simulation

In this section, to investigate the characteristics of the proposed law, several nonlinear simulations are performed.

### 6.1. Non-Maneuvering Moving Target

In this subsection, nonlinear simulations, with initial conditions shown in Table 2, are performed. From Table 2, $T_{\min} = 33.33$ s, so the desired impact times 35 s, 50 s, and 90 s are all available. From Table 3, $\alpha = 0.8$ is selected to eliminate errors within a fixed time. Further, based on the information provided in Table 3, $K$ is chosen as $K = 0.2$. Figure 3 shows the trajectories of the missile and the profiles of distance-to-go. It is evident from Figure 3 that in each case, the missile can reach the target at different desired impact times. Moreover, it is clearly shown in Figure 4a that the larger the initial impact time error, the greater the initial commanded acceleration will be.

**Table 2.** Parameter values used in simulation.

| Parameters | Values |
|---|---|
| Initial Missile Position | (0, 0) m |
| Missile Velocity | 300 m/s |
| The Bound on $|a_M|$ | 50 m/s$^2$ |
| Target Position | (10,000, 0) m |
| Initial Look Angle | 60° |
| Desired Impact Time | 35 s, 50 s, 90 s |

**Table 3.** Some useful values.

| $T_d$ | $\varepsilon_0$ | $\frac{\sqrt{3}}{9}\frac{2-\alpha}{1-\alpha}\frac{\pi}{T_d}$ | $\frac{1-\delta_0\cot\delta_0}{3|\varepsilon_0|}$ |
|---|---|---|---|
| 35 s | 5.3 s | 0.1036 | - |
| 50 s | −9.7 s | 0.0726 | 0.0136 |
| 90 s | −49.7 s | 0.0403 | 0.0027 |

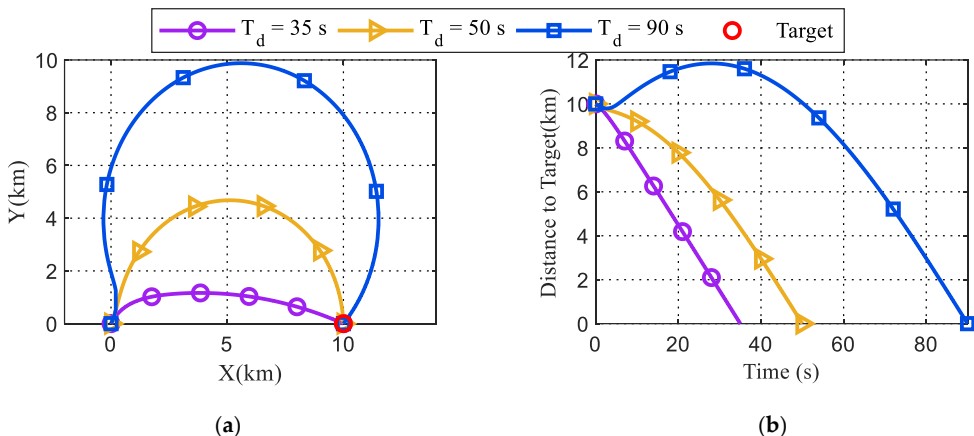

**Figure 3.** The results without look angle constraint. (**a**) Trajectories; (**b**) Distance to target.

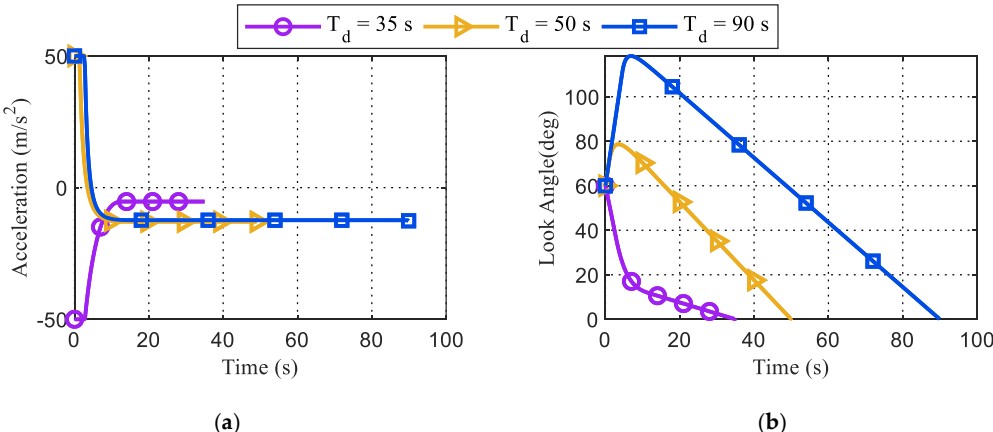

**Figure 4.** The results without look angle constraint. (**a**) Lateral acceleration; (**b**) Look angle.

When the impact time error $\varepsilon$ converges to zero within a finite time, the guidance command is only used to maintain a certain circular course, as is evident from Figure 4a. A striking feature of Figure 5 is that impact time errors always converge to zero within about 18 s, independent of the initial values, which is in accordance with the result provided by (59) As shown in Figure 4b, look angles for $T_d$ =35 s decrease with time owing to $\varepsilon_0 > 0$; but for cases where $T_d$ =50 s and $T_d$ =90 s, the curves show a trend of increasing and then decreasing due to $\varepsilon_0 < 0$. In the latter segment of the engagement, the missile will move along a circular course, causing the look angle to decrease to zero linearly, as shown in Figure 4b.

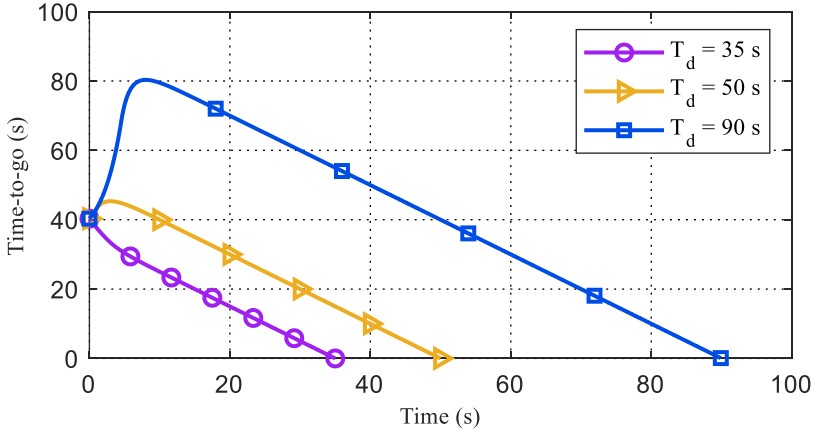

**Figure 5.** Time-to-go without look angle constraints.

In an overall sense, as described in Figure 6a, there is no significant difference between the two trajectories for $K = 0.2$ and $K = 0.4$ with $\alpha = 0.8$. However, the initial guidance command for $K = 0.4$ is greater than that for $K = 0.2$, according to (48). As a result, the profiles of the lateral accelerations with saturation show slight discrepancies, as seen in Figure 6b. As shown in Figure 7a, the value of $K$ has a very substantial effect on the settling time by revealing that a larger value of $K$ will expedite the convergence of impact time errors. Further, the settling time for $K = 0.4$ is approximately half of that for $K = 0.2$, which is parallel to (48). It is shown in Figure 6a that at the beginning of the engagement, the trajectory in the case $K = 0.4$ will have a greater curvature, which leads to a larger extrema value of the look angle, as seen in Figure 7b.

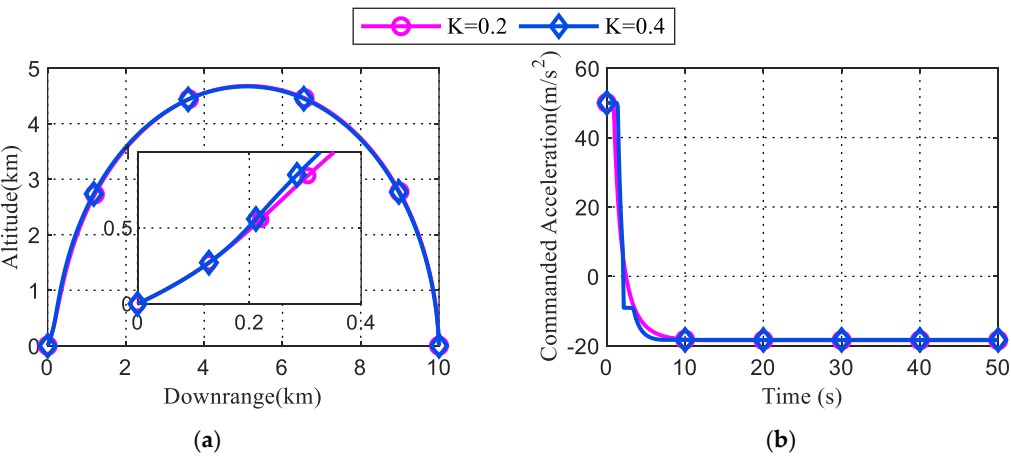

**Figure 6.** The results for different values of $K$ under $T_d = 50$ s. (**a**) Trajectories; (**b**) Lateral accelerations.

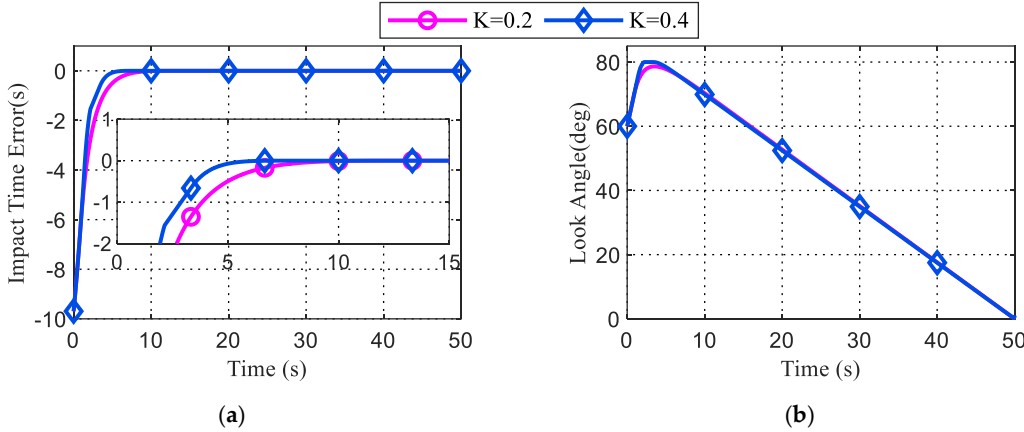

**Figure 7.** The results for different values of $K$ under $T_d = 50$ s. (**a**) Impact time errors; (**b**) Look angles.

### 6.2. Constrained Look Angle

It is necessary to choose a greater desired impact time to make the look angle reach the allowable values. As a result, $T_d = 90$ s is selected in this subsection. Other information is the same as that in Table 2. The allowed maximal look angles of the seeker are selected as $80°$, $90°$, and $100°$, respectively. Note that the allowed maximal look angle of $360°$ means there is no constraint on the look angles.

As shown in Figure 8a, when the look angle is considered, trajectories are quite different from the trajectories without limit on the look angle. Further, a smaller allowable maximal look angle will result in a trajectory with a lower altitude, but a greater scope in the downrange, which also occurs in [33].

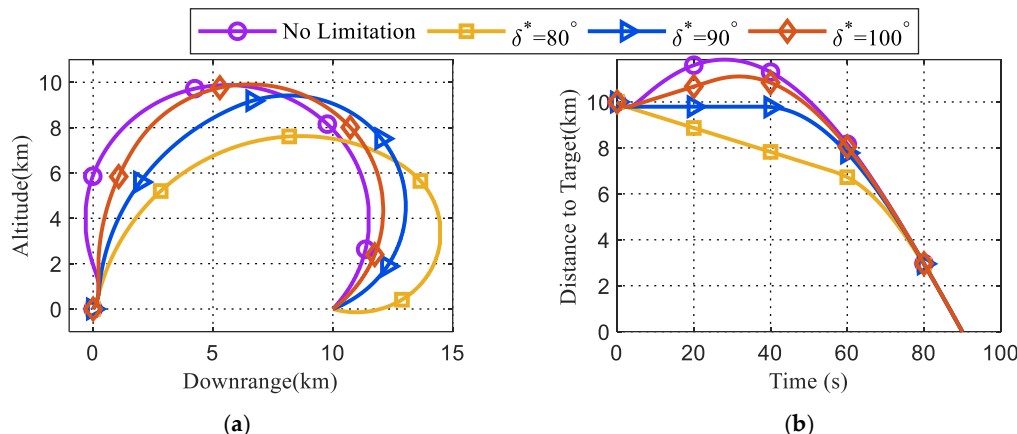

**Figure 8.** The results with different constraints on look angle. (**a**) Trajectories; (**b**) Distance-to-go.

For $\delta^* = 80°$, as can be seen from Figure 8b, the distance to target deceases in the whole engagement based on (2). Because the initial look angle is 60°, the look angle will increase to 80° due to $\varepsilon_0 > 0$ and then stay constant during DPP before decreasing with time in a linear pattern, as observed from Figure 9b. Note that, in case of $\delta^* = 80°$, the missile will hit the ground about 2 km away from the target, from which we can suggest that the combination of a longer impact time and a large look angle constraint is more reasonable. For $\delta^* = 90°$, the distance to target deceases at the beginning of the engagement before the look angle reaches 90°. Once the look angle reaches 90°, the guidance law is switched into DPP, where the distance to target and the look angle remain constant, as shown in Figures 8b and 9b. When $\Delta$, which is defined in (38), becomes positive, the guidance law is switched to FCITG, again leading to a decrease in look angles. For $\delta^* = 100°$, the behavior of the look angles is similar to the case $\delta^* = 90°$ as shown in Figure 9b. Meanwhile, the distance to target during DPP stage will increase due to look angles greater than 90°, as presented in Figure 8b.

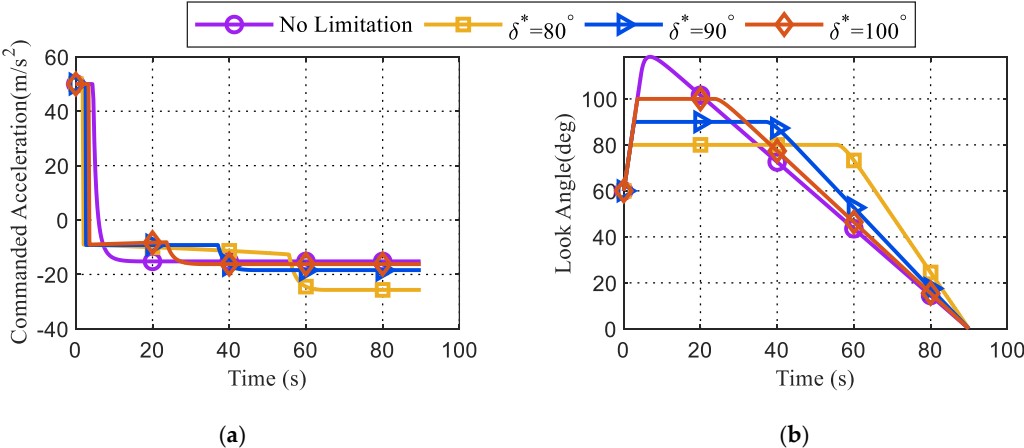

**Figure 9.** The results with different constraints on look angle. (**a**) Lateral accelerations; (**b**) Look angles.

In this subsection, the guidance can be divided into three stages: FCITG, DPP, and FCITG. Consequently, as depicted in Figure 9a, the lateral acceleration is piecewise in all three cases due to the switches between different guidance laws. Note that a jump of the guidance command only occurs at the first switch moment.

Now, different initial look angles will be investigated. The desired impact time $T_d = 50$ s is selected. The initial look angles are 1°, 30°, 60°, and 80°, respectively. The allowed maximal look angle is set to be $\delta^* = 80°$. Other information required is provided in Table 2. As shown in Figure 10a, different initial look angles can result in significant disparities in trajectories during the initial phase. In the later stages of the engagements,

these trajectories will show a similar pattern by following circular arcs after the impact time errors become zero, as seen in Figure 10a. The distance to target in each case decreases to zero monotonically, due to look angles no more than 80°, as shown in Figure 10b.

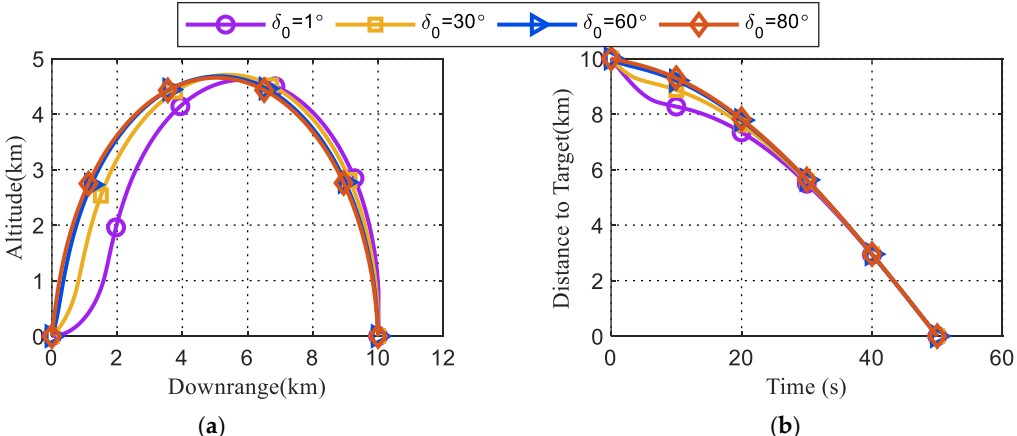

**Figure 10.** The results with different initial look angles under $\delta^* = 80°$. (**a**) Trajectories; (**b**) Distance to target.

In case of $\delta_0 = 1°$, Figure 11a shows that the profile of the lateral accelerations is piecewise with a jump because the look angle will reach the allowed maximal value leading to a switch between FCITG and DPP, while the profiles of the lateral accelerations in the other three cases do not suffer from a sudden change. In summary, FCITG is insensitive to the values of the initial look angles.

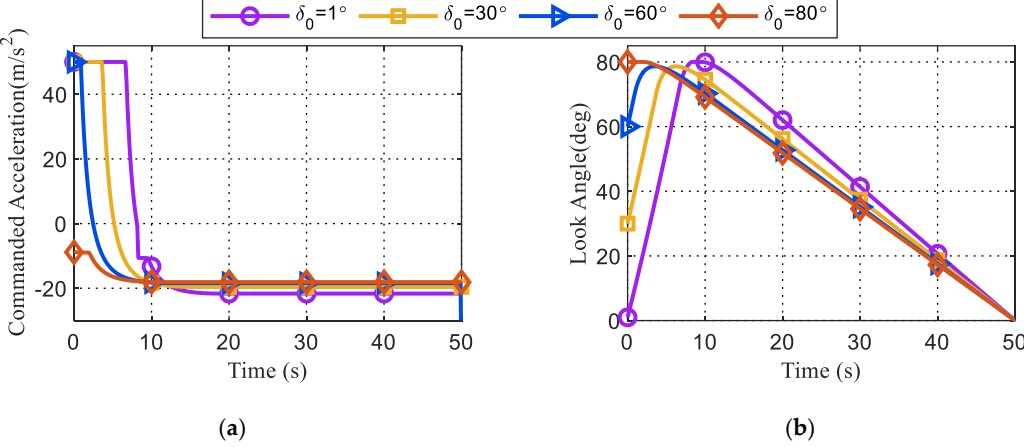

**Figure 11.** The results with different initial look angles under $\delta^* = 80°$. (**a**) Lateral accelerations; (**b**) Look angles.

### 6.3. Salvo Attack Scenario

In this subsection, a simulation of a salvo attack in which three missiles engage a stationary target is performed. The target position is set as (10,000, 0) m. Missiles, which are denoted by M1, M2, and M3, are launched from different locations with different initial look angles and different speeds. To better verify the effectiveness of the algorithm, the upper limit of the command acceleration is also set to different values. The detailed information for the salvo attack is summarized in Table 4. To be more realistic, constraints on look angles are also considered by using 80° as the allowed value in the simulation.

**Table 4.** Engagement conditions.

| Missiles | Initial Missile Position | Initial Look Angle | Missile Speed | The Bound on $|a_M|$ |
|---|---|---|---|---|
| M1 | (0, 0) m | 60° | 300 m/s | 50 m/s² |
| M2 | (4000, 0) m | 40° | 250 m/s | 40 m/s² |
| M3 | (−4000, 0) m | 20° | 350 m/s | 60 m/s² |

As seen from Figure 12a, even if the missiles are launched with initial ranges of great discrepancy, from 6000 m to 14,000 m, each of missiles can hit the target at $T_d = 50$ s, while in [1], one prerequisite to achieve a simultaneous attack is that missiles should be launched with similar ranges from the target. Moreover, as shown in Figure 12b, the three curves have different gradients in the region just before interception at $T_d = 50$ s due to a different closing speed of each missile.

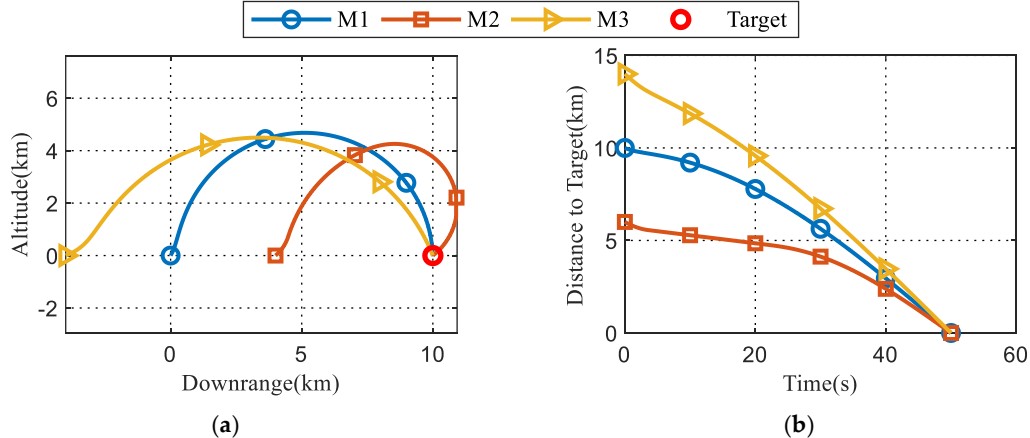

(**a**)

(**b**)

**Figure 12.** The results in a salvo attack. (**a**) Trajectories; (**b**) Distance to target.

As can be seen from Figure 13, when the maximal look angle is less than the allowable maximal value for M1 and M3, DPP is not employed. While for M2, when the look angle reaches 80°, DPP is then applied. Accordingly, only the guidance command for M2 shows a jump, as can be seen in Figure 14a. Most importantly, impact time errors all converge to zero within about 25 s, regardless of initial impact errors, as observed in Figure 13b.

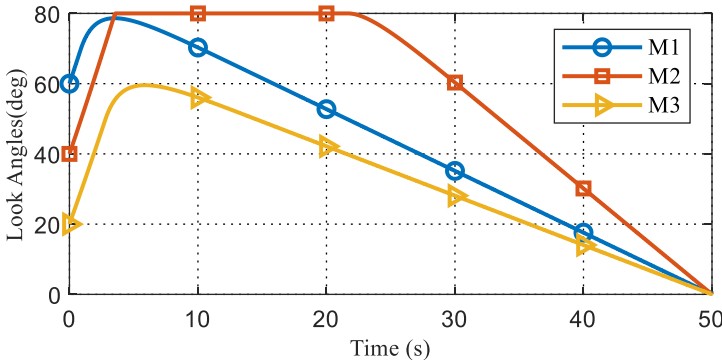

**Figure 13.** Look angles in a salvo attack.

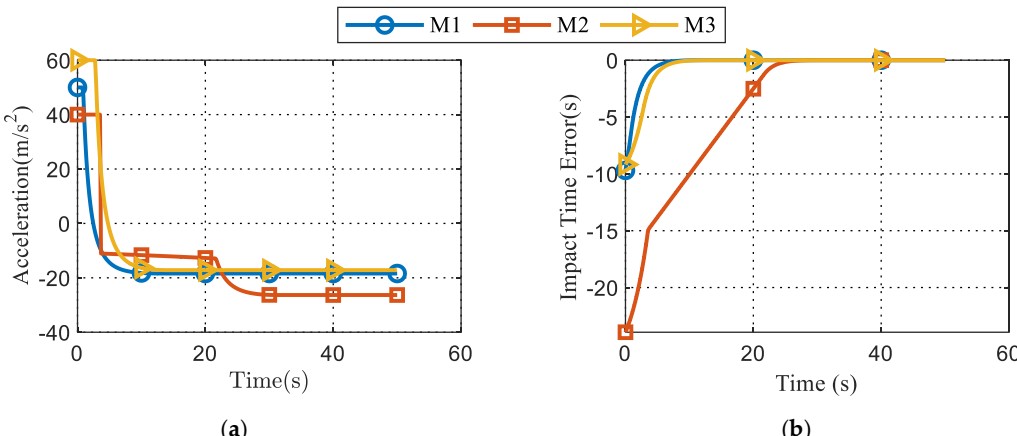

(a)    (b)

**Figure 14.** The results in a salvo attack. (**a**) Lateral accelerations; (**b**) Impact time errors.

*6.4. Extension to a Moving Target*

For a moving target, the parameter values are provided in Table 5. As can be seen from Figure 15, the missile can intercept the target at different desired moments under different initial conditions by aiming for PIP. As seen from Figure 16b, this simulation is only to verify the effectiveness of the proposed algorithm, so the look angle constraint is not considered. As a result, the guidance command is continuous as shown in Figure 16a. In fact, the look angle constraint can be achieved by switching the guidance law.

**Table 5.** Parameter values for a moving target.

| Parameters | Values |
| --- | --- |
| Initial Missile Position | (0, 0) m |
| Missile Speed | 300 m/s |
| The Bound on guidance command | 50 m/s$^2$ |
| Initial Target Position | (10,000, 0) m |
| Initial Look Angle | 30°, 45°, 60° |
| Desired Impact Time | 50 s, 70 s, 90 s |
| Target Speed | 50 m/s |
| Target Path Angle | 0° |

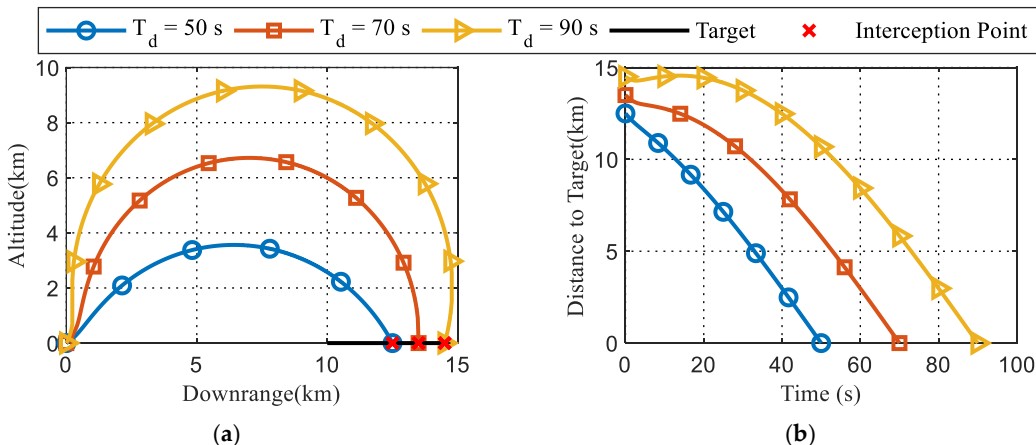

(a)    (b)

**Figure 15.** The results for a moving target. (**a**) Trajectories; (**b**) Distance to target.

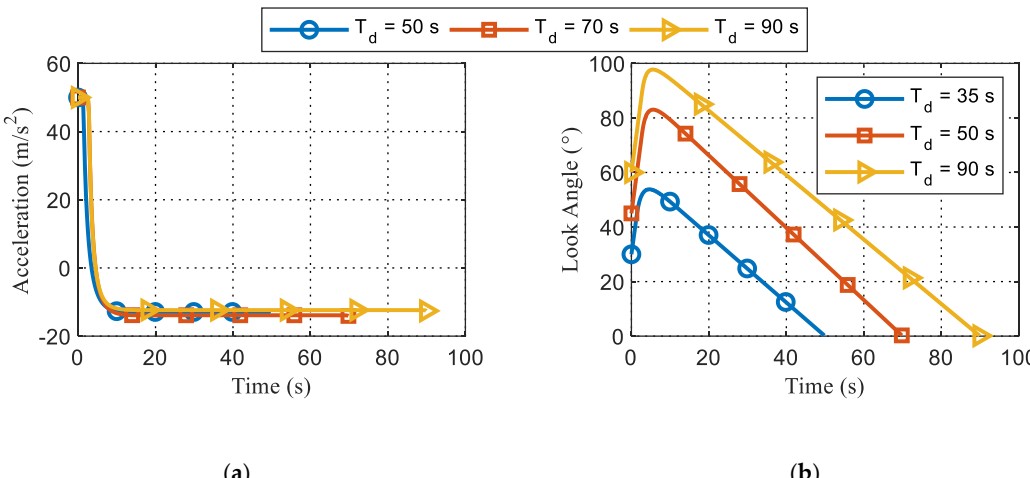

**Figure 16.** The results for a moving target. (**a**) Command acceleration; (**b**) Look angle.

### 6.5. Comparative Study

Based on the parameter values in Table 2 with $\delta^* = 80°$, each desired impact time between 35 s and 90 s, with a 1-second interval, is simulated to verify the effectiveness of the FCITG. Figure 17a shows that $T_d \in [35, 90]$ s can be achievable by FCITG.

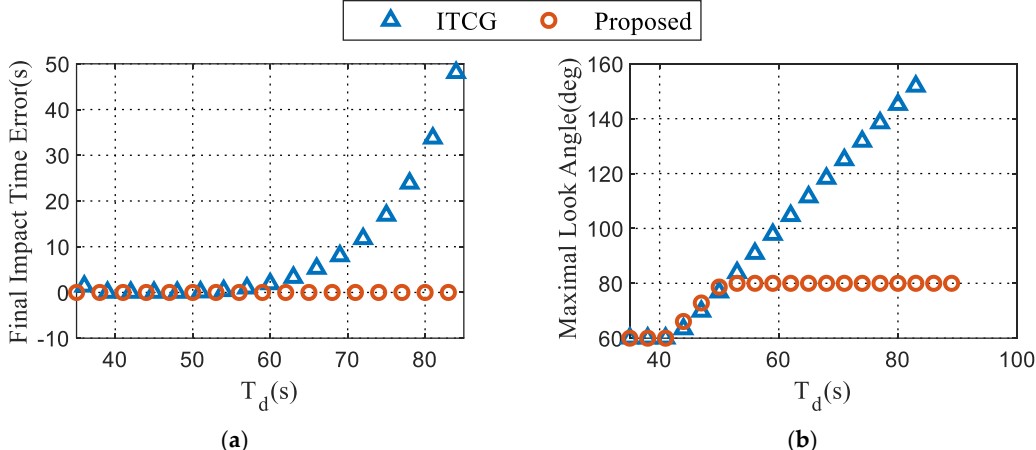

**Figure 17.** The comparison results. (**a**) Final impact time error; (**b**) Maximal look angle.

Linearization and small angle assumptions are not involved in the derivation of the proposed guidance law, so the value of $T_d$ has almost no influence on the final impact time error. By introducing DPP during the middle phase of the guidance in cases where $T_d$ is great than about 50 s, constraints on look angles can be satisfied, as observed from Figure 17b.

From the parameter values in Table 2, the flight time governed by PNG can be obtained as $t_f^{PNG} = 37.36$ s, with navigation constant being 3, by simulations or numerical computations. This is also one of the reasons why the lower bound of the desired impact time is chosen as 35 s in this subsection. It is clearly observed from Figure 17a that only the desired impact time between about 38 s and 50 s can be achievable using the ITCG presented in [1]. The longer the desired impact time, the greater the maximal look angle. When the desired impact time is greater than 50 s, the small angle approximation, which was a precondition for the estimation of time-to-go in [1], can produce large errors. Besides, the constraint on the look angle ITCG in [1] was not considered. As a result, when the desired impact time becomes large enough, there will be a dramatic increase in both the final impact time error and the maximal look angle, as can be seen from Figure 17. This

means that the proposed FCITG can be applied to a much wider range of the achievable desired impact time.

To measure the energy cost during the guidance, an energy-related index is defined as $J = \int_0^{T_d} |a_M| \mathrm{d}t$. The comprehensive performance of the proposed FCITG will be evaluated from four aspects, including energy cost, miss distance, final impact time error, and the maximal look angle. In fairness, as in [1], the navigation constant is chosen as $N = 3$, and the PI controller gains are chosen as $K_P = -150$, and $K_I = -15$, according to [33]. As seen in Table 6, both the proposed FCITG and CITG in [33] can reach the target at $T_d = 35$ s, with look angles no more than $60°$ throughout the whole engagement, while the ITCG in [1] can generate a final impact time error up to 2.36 s. Besides, compared with the other two guidance laws, the energy cost is the least for the proposed FCITG.

**Table 6.** Characteristics for $T_d = 35$ s.

| Guidance Law | Energy Cost | Final Impact Time Error | Max Look Angle |
|---|---|---|---|
| Proposed | 425.7 m/s | 0.0001 s | 60° |
| ITCG in Ref. [1] | 471.3 m/s | 2.36 s | 60° |
| CITG in Ref. [33] | 433.4 m/s | 0.0008 s | 60° |

For $T_d = 50$ s, both the proposed FCITG and ITCG in [1] can hit the target at the desired impact time without violating the constraint on the look angle, as shown in Table 7. Since ITCG in [1] is an energy-optimal based guidance law, its energy cost is less than that in the proposed FCITG. Although $T_d = 50$ s is also achievable by utilizing the CITG in [33], the maximal look angle contradicts with the constraint on the look angle because of the use of a PI controller, which can make the seeker fail to detect the target. It is well known that a PID controller can produce oscillations, or an overshoot, by employing improper gains. However, finding proper gains for a PID controller may be time-consuming and experience-dependent, while the parameter values in FCITG can be determined by rules given by Table 1. The final impact time error is a particularly large, as seen in Figure 17, results yielded by the ITCG in [1] are no longer displayed in Table 8, and because the trajectory using the ITCG in [1] is strange, as shown in Figure 18a. An overshoot in look angles arises again by using the CITG in [33] when it comes to the case of d $T_d = 90$ s, as observed in Figure 19a and Table 8.

**Table 7.** Characteristics for $T_d = 50$ s.

| Guidance Law | Energy Cost | Final Impact Time Error | Max Look Angle |
|---|---|---|---|
| Proposed | 917.1 m/s | 0.0001 s | 78.64° |
| ITCG in Ref. [1] | 710.3 m/s | 0.04 s | 76.79° |
| CITG in Ref. [33] | 931.8 m/s | 0.0009 s | 81.33° |

**Table 8.** Characteristics for $T_d = 90$ s.

| Guidance Law | Energy Cost | Final Impact Time Error | Max Look Angle |
|---|---|---|---|
| Proposed | 1521.7 m/s | 0.0001 s | 80° |
| ITCG in Ref. [1] | - | - | - |
| CITG in Ref. [33] | 1463.2 m/s | 0.0009 s | 84.6° |

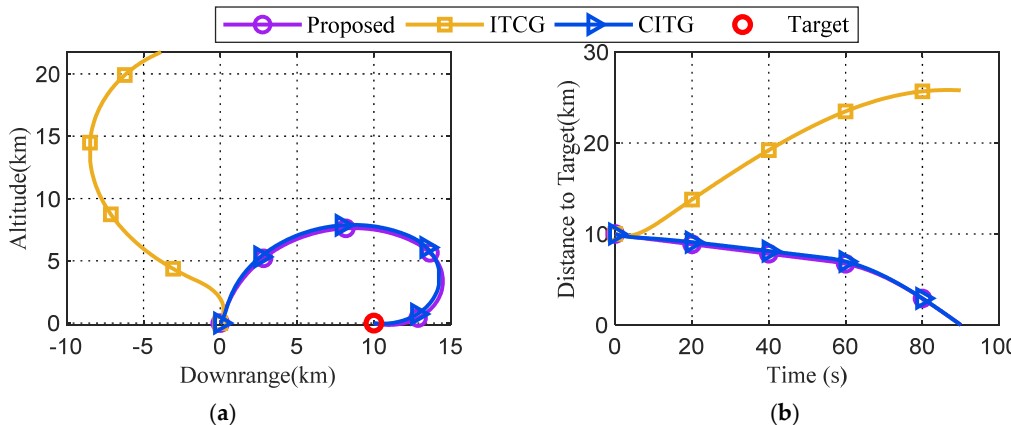

**Figure 18.** The results for different guidance laws for $T_d = 90$ s. (**a**) Trajectories; (**b**) Distance to target.

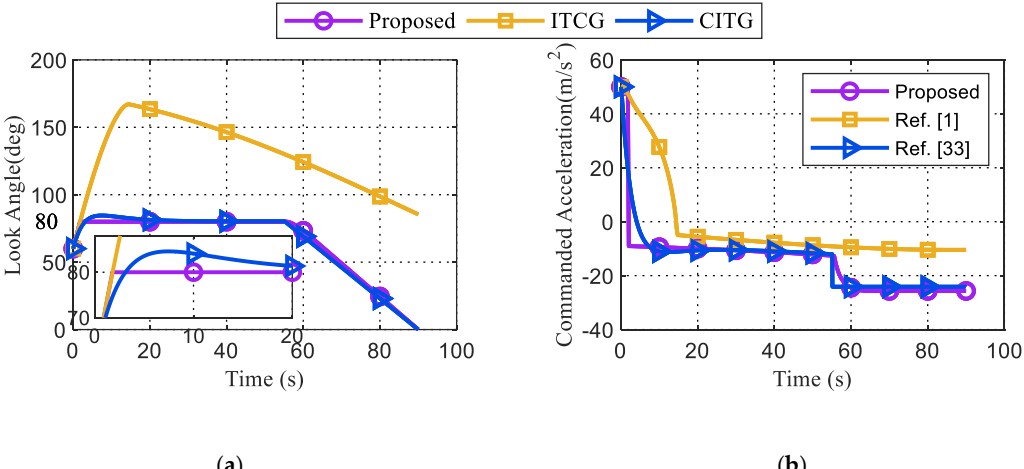

**Figure 19.** The results for different guidance laws for $T_d = 90$ s. (**a**) Look angles; (**b**) Lateral accelerations.

In general, simulations have shown that the proposed FCITG in this paper can hit a stationary or a moving, but non-maneuvering, target at a desired impact time, with zero miss distance. Further, the energy consumption is appropriate under different initial conditions, without violating the look angle constraint.

## 7. Conclusions

This paper proposes a new approach, called FCITG, to control impact time based on circular guidance and the fixed-time stability theory. Compared with the previous representative guidance laws, the proposed guidance law was derived based on exact time-to-go computation, instead of the estimation of time-to-go. Moreover, the criteria for the selection of parameter values, which can be used to determine the upper limit on the settling time, are established by rigorous derivations and justified by numerical simulations. The nonlinear framework used for the proposed guidance law makes it applicable, even for engagements with large heading errors. The derived guidance law is shown to satisfy a wide range of desired impact times. Furthermore, the desired impact time can be even less than the initial time-to-go estimate, which may be not feasible using many existing guidance laws. Next, the look angle constraint is addressed through introducing DPP, when necessary. By employing the FCITG proposed and sharing information on the desired impact time, a group of missiles can realize a salvo attack for a moving target, as well as a stationary target. The reliability of the salvo attack strategy can be guaranteed by the fixed-time convergence of the impact time errors. Simulations have shown that the proposed guidance law was not sensitive to heading errors, initial positions, and the expected initial

flight time. Future work will focus on the constraints on both impact time and impact angle, as well as 3-D scenarios.

**Author Contributions:** Conceptualization, X.L.; validation, X.L.; writing—original draft preparation, X.L.; writing—review and editing, Z.C., W.C., T.W. and H.S. All authors have read and agreed to the published version of the manuscript.

**Funding:** This research was funded by the China Postdoctoral Science Foundation (Grant No. 2021M700321).

**Institutional Review Board Statement:** Not applicable.

**Informed Consent Statement:** Not applicable.

**Data Availability Statement:** The data used to support the findings of this study are available from the corresponding author upon request.

**Conflicts of Interest:** The authors declare no conflict of interest.

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
