# Peer review of "Fixed-Time Circular Impact-Time Guidance with Look Angle Constraint"

_aerospace, doi:10.3390/aerospace9070356_

Round 1
Reviewer 1 Report
In general, the paper submitted for review entitled „Fixed-time Circular Impact-time Guidance with Look Angle Constraint” I find as interesting voice in the discussion of issues related to modern methods of guiding missiles. The work provides theoretical basis for the development of algorithms for the synchronized missile guidance to ground and surface targets. It should be emphasized a very thorough literature review and a clear indication of the autors' contribution to the subject matter. Nevertheless, a few issues raise my doubts, hence I kindly ask the authors to comment on the issues included as attachement.

Reviewer 2 Report
The paper tilted “Fixed-time Circular Impact-time Guidance with Look Angle 2 Constraint” describes a fixed-time nonlinear circular guidance law that satisfies the impact time constraint. This paper proposes a new and interesting approach to control impact time based on circular guidance and the fixed-time stability theory. It should be emphesized, that compared with the previous representative guidance laws, the proposed guidance law was derived based on exact time to-go computation instead of estimation of time-to-go. You should expect that future work will focus on the constraints on both impact time and impact angle as well as 3-D scenarios.
The article is written at a good level, both a meritoric as an editorial level. The topic is up to date.
Reviewer 3 Report
comments attached

Round 2
Reviewer 1 Report
In my opinion, the paper can be published in present form.
Reviewer 3 Report
Point 8 of my review was not answered satisfactorily